# Developing an intervention to improve early infant HIV diagnosis service uptake among postpartum women in Malawi's primary healthcare using a co-designing approach with stakeholders

Leticia Chimwemwe Suwedi-Kapesa[1,2,3,4]*, Augustine Talumba Choko[1,2], Alinane Linda Nyondo-Mipando[5,6], Jenifer Hezekiah Zimba[7], Edda Lipipa[7], Dorcus Nothale[7], Afunawo Mdala Maulukira[8], Joe Nkhonjera[9], Melody Sakala[3], Nicola Desmond[1,10], Angela Obasi[1,11]

1 Department of International Public Health, Liverpool School of Tropical Medicine, Liverpool, United Kingdom, 2 Public Health Research Group, Malawi-Liverpool-Wellcome Trust Clinical Research Programme, Blantyre, Malawi, 3 Policy Unit, Malawi-Liverpool-Wellcome Trust Clinical Research Programme, Blantyre, Malawi, 4 Department of Public Health Surveillance and Disease Intelligence, Public Health Institute of Malawi, Lilongwe, Malawi, 5 Department of Women's and Children's Health, University of Liverpool, Liverpool, United Kingdom, 6 Department of Health Systems and Policy, Kamuzu University of Health Sciences, Blantyre, Malawi, 7 Department of Nursing, Blantyre District Health Office, Blantyre, Malawi, 8 Department of Technical Community ART Dispensation, Elizabeth Glaser Paediatric AIDS Foundation, Lilongwe, Malawi, 9 Directorate of HIV, STI, and Viral Hepatitis, Ministry of Health, Lilongwe, Malawi, 10 Department of Global Health and Development, London School of Hygiene and Tropical Medicine, London, United Kingdom, 11 Axess Sexual Health, Liverpool University Hospitals NHS Foundation Trust, Liverpool, United Kingdom

* lsuwedi@mlw.mw

## Abstract

Low health service use by women and infants after birth limits early infant HIV diagnosis (EID). From August 2021 to December 2022, we collaborated with 44 healthcare workers (HCW), service users, and non-governmental organisation stakeholders from seven public facilities and five non-governmental organisations in Blantyre, building on a previous study. We analysed context-specific problems in EID services and co-designed a context-appropriate enhanced health system intervention to improve the uptake of six weeks' EID services in primary health facilities in Blantyre, Malawi, using qualitative methods and co-designing workshops. The Behaviour Change Wheel, Theoretical Domain Framework and Consolidated Framework for sustainability constructs in healthcare guided the work-shops. Reflexive thematic analysis of the data showed that stakeholders found that EID services were sub-optimal and identified challenges to service provision in 5 key areas: (1) client identification, (2) context-appropriate client-centred service integration, (3) HCW coordination and accountability, (4) HCW capacity building for optimal service delivery, and (5) intervention sustainability. Specifically, client and HCW stigma perceptions, referral gaps, resource challenges, HCW lack of time and poor documentation affected client identification; HCW clustered work shifts to extend off-duty periods, failure to synchronise client appointments, and lack of resources were barriers to client-centred integrated services;

**Data availability statement:** All relevant data are within the paper and its Supporting Information files.

**Funding:** Leticia Chimwemwe Suwedi-Kapesa (LCSK) received funding from the following: (i) Commonwealth Scholarship Commission (CSC) grant number (MWCS-2020-286) (ii) Institutional Translation Partnership award (Wellcome) through the Policy Catalyst Fund—Malawi Liverpool Wellcome Trust (206545/Z/17/Z). The policy catalyst fund supported the stakeholder engagement workshops. The funders had no role in study design, data collection and analysis, publication decisions, or manuscript preparation. The content does not represent the views of the funders.

**Competing interests:** The authors have declared that no competing interests exist.

dysfunctional teams, minimal supervision and misconduct among HCW impacted coordination and accountability; and lack of information sharing and limited training reduced HCW capacity for service delivery. Context-appropriate stakeholder informed co-design initiatives to address identified challenges included: clients' unique identifiers, booking systems, strengthening leadership, data validation, care pathways, and facility-based training. We recommend evaluating these initiatives in low resource settings as they have potential to address the identified EID service implementation gaps and significantly improve the EID of HIV in contexts of greatest need.

## Introduction

Vertical transmission of HIV rates during the breastfeeding period are high and are a threat to achieving Sustainable Development Goal 3.3 of ending the AIDS pandemic by 2030 [1,2]. In 2022, vertical transmission of HIV rate was significantly higher than the 5% global target at 11% globally and 7.9% in Malawi [3,4]. The high rate of vertical transmission of HIV is due to a combination of breastfeeding women disengaging from anti-retroviral therapy (ART) and new maternal HIV infections [5–7]. Retention of women in postpartum care during breastfeeding facilitates early ART initiation, promotes maternal health and prevents vertical transmission of HIV [2]. Maternal retention in both postpartum and ART care is critical to facilitate ART prophylaxis to infants exposed to HIV; to early infant diagnosis of HIV (EID); and to timely provision of appropriate treatment and support to prevent high mortality among infants living with HIV [8]. Postpartum and ART care retention is critical but remains suboptimal to reach the third 95-95-95 target for viral suppression. Postnatal utilisation coverage at six weeks is 52.5% in sub-Saharan Africa and 48.4% in Malawi [9,10]. ART retention is 72.3% and 82%, respectively [5,11].

Health system factors such as poor healthcare workers (HCW) attitude, lack of service integration including unsynchronised appointments, long turnaround of HIV test results and a lack of HIV diagnosis contribute to women disengaging from care [2,12–14]. Potential strategies to improve retention in care and HIV testing of infants exposed to HIV include integrating services for the mother and infant and providing point-of-care (POC) HIV testing [2,15].

According to the World Health Organisation (WHO) 2021, children under 18 months, at 4-6 weeks and 9 months, should be offered POC Deoxyribonucleic Acid Polymerase Chain Reaction HIV testing [8]. The global goal is to achieve 95% testing for infants exposed to HIV at six weeks, yet only 67% were tested in 2023 [7,16]. In Malawi EID uptake at six weeks increased from 40% in 2020 to 85% in 2023. Since 2018, Malawi has implemented POC EID testing at six weeks in limited settings. In 2022, Malawi adapted WHO recommendations and enhanced its offering of point-of-care (POC) testing at six weeks by designating facilities with POC machines as hubs to process samples from facilities without POC machines, a strategy referred to as near point-of-care testing [17]. Suwedi-Kapesa et al. evaluation of EID services [18] found suboptimal six-week EID uptake in two primary healthcare facilities in Blantyre, Malawi. Only 39 (32%) of 163 infants exposed to HIV born at the sites in 2018 were enrolled in HIV care at birth, and 85 (52%) were tested at six weeks, of whom 6 (8%) lacked results. From 2019 to 2020, there were delays and implementation gaps in EID services (up to six weeks) due to process, capacity, and system-level factors, such as HCW's failure to identify women with infants exposed to HIV, limited capacity of POC machines, and challenges with service integration in Blantyre, Malawi [18]. These findings highlighted the need to develop an enhanced health system intervention (EEH) to improve service uptake if WHO guidelines

are to be met in this setting. This paper builds on the Suwedi-Kapesa et al. evaluation of EID services to co-design the EEH intervention.

We utilised a framework for developing and evaluating complex interventions based on the updated guidelines from the Medical Research Council (MRC) [19]. The MRC offers comprehensive factors to prompt researchers to reflect on the system, consider context, stakeholders, theories, key uncertainties, intervention refinement, and economics, and develop context-responsive interventions. Our focus centred on context, involving stakeholders, refining interventions, and integrating a theory. This approach enabled us to validate the evaluation of EID services, examine context-specific challenges, and co-design an enhanced, context-appropriate health system intervention to improve uptake of EID services at six weeks.

## Materials and methods

### Ethics statement

The study was reviewed and approved by the College of Medicine Research & Ethics Committee (P.04/22/3607) and the Liverpool School of Tropical Medicine Research Ethics Committee (22-025). Participants provided informed verbal consent to attend the workshops, which involved training on the frameworks, validating the findings of Suwedi-Kapesa et al., and developing and refining the intervention guided by frameworks. We first sought written approval from the health and social services director to release the mapped HCW stakeholder participants. The non-governmental organisations gave us contacts to approach those who represented them. Participants were given the invitation letter and study information. After two days, the researcher followed up by phone if they had questions and were interested in participating in the workshops. Interested participants opted in, while two opted out. Participants who opted in were sent the workshop materials and agenda by phone (WhatsApp) two weeks before the workshops.

### Inclusivity in global research

Additional information regarding the ethical, cultural and scientific considerations specific to inclusivity in global research is included in the supporting information (S1 Text)

### Adapted theory

To steer intervention development, we applied multiple measures, including an emphasis on context, involved stakeholders integrated a theory and refined intervention as core components of the MRC. We conducted stakeholder mapping to identify stakeholders in EID services and used two workshops to engage them in understanding the context of EID services. Potential HIV stakeholders were listed, including service users, non-governmental organisations and HCW involved in infants' and women's health at different levels and locations within the Blantyre District health system. A power interest matrix was used to categorise stakeholders we engaged at various stages of the co-designing process [20,21] to capture the relationship between their power and interest position to guide stakeholder selection for the workshops (Fig 1) [22].

We adapted the Behaviour Change Wheel (BCW) with its capability, opportunity, motivation-behaviour (COM-B) intervention design constructs, the Theoretical Domains Framework, and the Consolidated Framework for Sustainability Constructs in Healthcare) as our theory to guide the co-designing process to develop and modify the intervention [19,23–25] (Fig 2). Using theories guides the collaboration process in developing interventions and clarifies the complex behaviour changes needed to achieve goals and interventions [26].

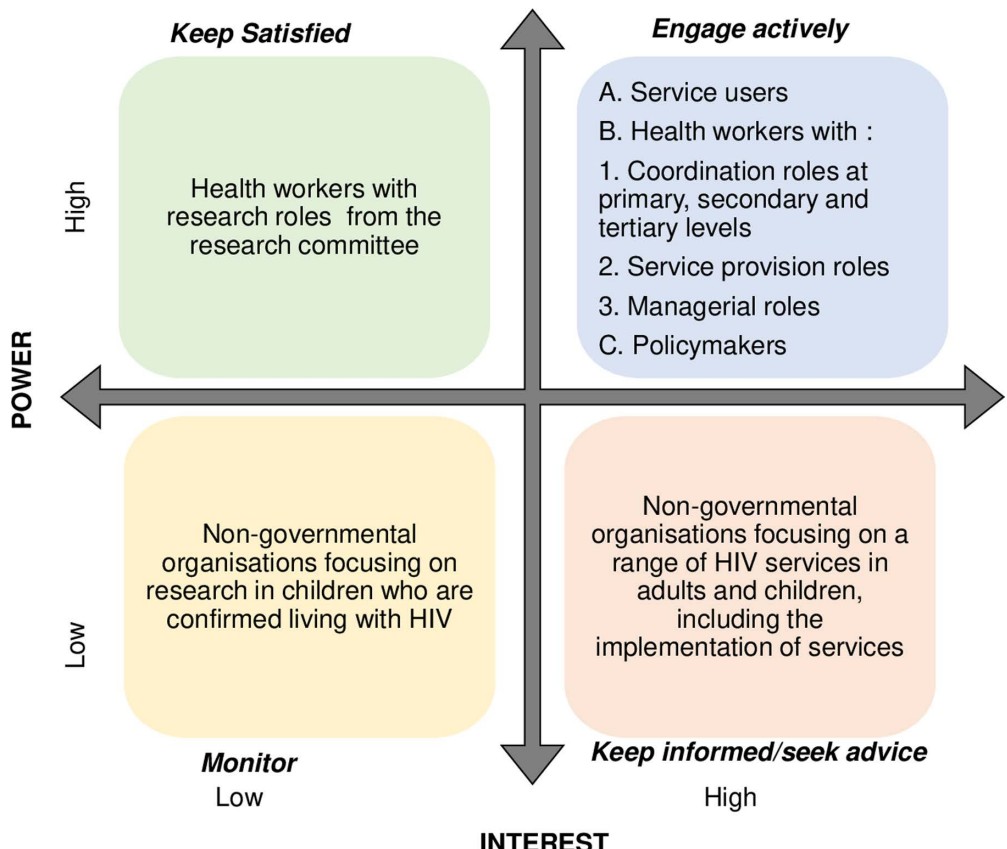

**Fig 1. Stakeholder mapping matrix.**

The BCW states that physical and psychological capability, social and physical opportunity, and reflective and automatic motivation are required to adopt the desired behaviour [25,27]. The Theoretical Domains Framework (TDF) complements the BCW to describe determinants of behaviour [28,29]. The BCW and TDF provide three stages to formulate a behaviour change approach [30,31]. Stage 1 defines the problem, selects target behaviour, specifies target behaviour and identifies what needs to change. Stage 2 identifies intervention type and policy categories, while stage 3 identifies behaviour change techniques and modes of delivery [29–32]. The consolidated Framework for Sustainability Constructs in Healthcare was included to ensure the inclusion of planning for intervention sustainability [24]. Overall, as shown in (Fig 2). The adapted theory guided the co-designing of the intervention in three phases.

Phase 1: Stakeholder identification involved creating a list of all stakeholders in EID services within the Blantyre district, assessing their potential influence and power, and organising invitations for participation workshops.

Phase 2: Stakeholder-led EEH intervention development at formative workshop 1; This aimed at engaging stakeholders to identify and define priority problems, specify target behaviours to address, determine required changes, and identify who could perform target behaviours, along with where to change and the timing, guided by the BCW and TDF frameworks (Fig 2) as the main topics for discussion [28]. LCSK presented findings from Suwedi-Kapesa et al.'s evaluation of EID service (S2 Text) [22]. Through group discussions, stakeholders first reflected on and validated these findings. LCSK then presented an overview of initiatives to enhance EID services in Africa; for example, using instant text messaging

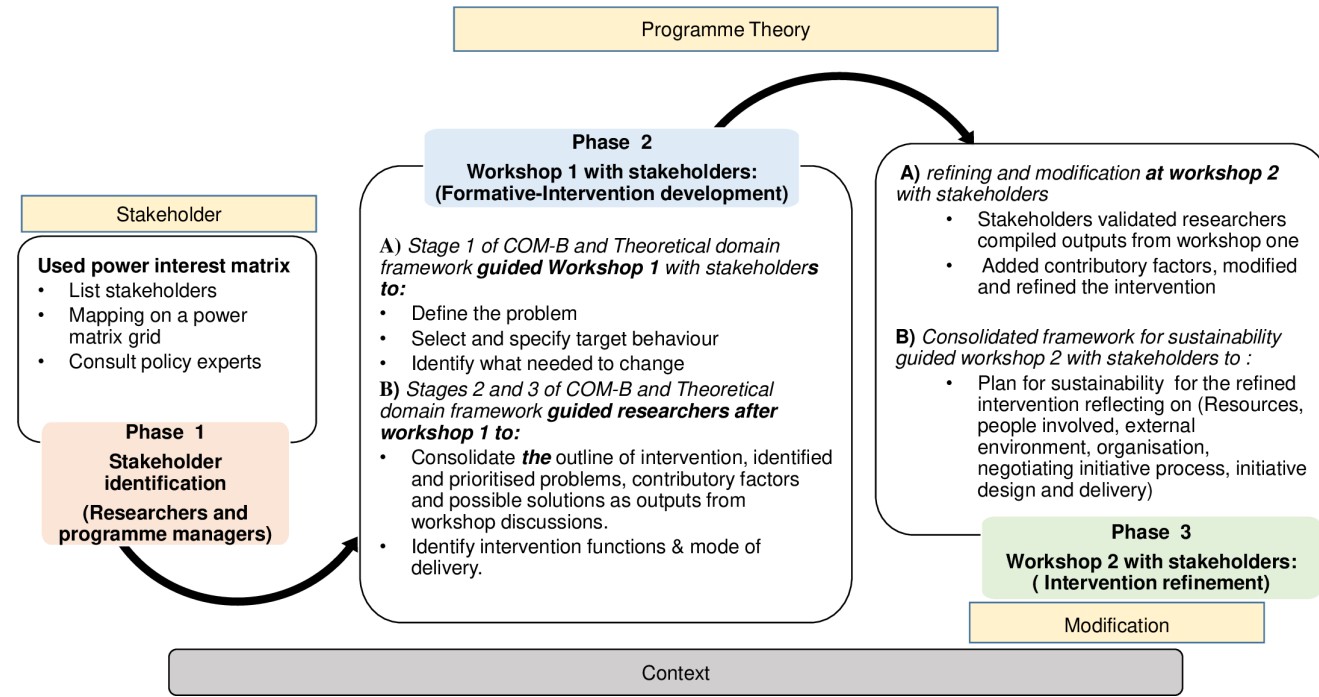

**Fig 2. Adapted theory.**

(SMS) to remind women with IEH of appointment dates [20,39–41]. To build the capacity of stakeholders, the researcher explained the BCW and TDF intervention development stage one that guided follow-up discussions. After the workshop, LCSK consolidated workshop 1 findings (S1 Table, S3 Text), guided by stages two and three of BCW and TDF.

Phase 3: Stakeholder-led EEH intervention refinement and sustainability measures at workshops 2. This involved engaging stakeholders to refine the EEH intervention's outline, discuss its feasibility, clarify content, explore uncertainties, adapt to changes in context, and plan for sustainability, guided by the Consolidated Framework for Sustainability Constructs in Healthcare as the discussion topic. LCSK presented the outline of the EEH intervention and identified and prioritised problems along with their contributory factors from workshop one, which stakeholders validated. This was followed by discussions and refinements of each initiative, considering the workshop's aims, stakeholders' experiences, contributory factors to the problems, and the context. A booking register and additional pages for available data sources were designed to accommodate intervention initiatives and the new clinical management of HIV in children and adults (2022) guidelines. Finally, stakeholders discussed sustaining the developed interventions (Fig 2).

## Study setting

The study was conducted in Blantyre, the second-largest city in Malawi, which had a high HIV prevalence of 14.2% in the general population and 10.9% among pregnant women [33]. By 2022, Blantyre had 28 government facilities (9 urban, 19 rural). Blantyre had 53 certified sites offering EID, including private and non-governmental organisations services. Four government facilities provided POC HIV testing for six weeks using Abbott mPima machines that processed one sample in 54 minutes. Four sites had Cepheid GeneXpert (1 with 16 modules processing 16 samples in 60-90 minutes and 3 with four modules processing four samples at a

time. The Cepheid GeneXpert machines were primarily used for TB testing, integrating with EID and Viral load [17]. One facility with 16 modules, Cepheid GeneXpert, processed samples from all sites without POC machines as a hub. It took four weeks for the facilities without POC to receive results from the hub. The hub facility was in Blantyre urban, 5 kilometres from where previously centralised testing was conducted.

We co-designed the intervention to be evaluated in one urban and one rural setting - building on Suwedi-Kapesa et al. evaluation study sites based on 1) the location of government facilities, urban and rural. 2) High and low catchment population (urban: 143,515 as second largest primary facility and rural: 10,069). 3) Presence or absence of POC machines (Urban: with POC machine and rural: without).

The co-designing intervention workshops included stakeholders from Suwedi-Kapesa et al.'s study settings and additional sites, totalling three urban, two rural primary health facilities, the district health office, Queen Elizabeth Central Hospital (QECH, the tertiary referral hospital to which primary facilities referred their clients) and five non-governmental organisations. Additional urban and rural facilities were chosen based on similar characteristics to the Suwedi-Kapesa et al—study setting (S2 Table) to ensure diverse perspectives and consideration for scaling up. The workshops occurred in a neutral setting in a conference room for Eyesight QECH [25]. Overall, the study settings offered insights into the context of implementing EID services in primary healthcare in Blantyre. All primary government facilities offered free public services comprising family planning, outpatient facilities, child growth monitoring, antenatal, immunisation, delivery, and HIV services for adults and infants. Unlike rural facilities, urban facilities have HCW with advanced training and skills, including degree-qualified nurses [18].

## Study design

We used a descriptive study design for the stakeholder analysis and a qualitative study design for the stakeholder discussion.

## Recruitment and enrolment of stakeholders

Before each workshop, Blantyre managers (JHZ & EL) and the principal researcher (LCSK) selected participants through stakeholder mapping as described in the adapted theory. We shared the list of potential Ministry of Health policy and non-governmental stakeholders with policy engagement experts at the Malawi Liverpool Wellcome Trust to review the stakeholder mapping process and identify any key stakeholders we may have missed. The HCW stakeholders included nurses, laboratory officers, HIV diagnostic assistants, health surveillance assistants, managers with coordination roles, and senior managers. The HCW identified maternal service users aged over 18 years who had received more than six months of mother-infant care at their facility. All participants were compensated for their transport costs based on submitted receipts.

## Data collection

We collected descriptive demographic characteristics of stakeholders and qualitative data on the discussions during the two workshops held between August 2021 (formative) and December 2022 (refinement/modification). Each workshop lasted two days, with an average duration of six hours each day. Discussions were conducted in Chichewa, the local language, within small homogeneous groups purposively created to address power dynamics and were later presented to the larger group for further discussion. There were five groups: non-governmental organisations, service users, service providers, coordinators, and managers,

with participant numbers ranging from two to six in each group. Each group self-appointed a facilitator to guide the discussions on designated topics (as described in phases 2 and 3 in the adapted theory), empowering them to take ownership of the discussions and document outcomes on flip charts. LCSK, JHZ, and EL, oriented to the study and experienced in moderating workshops in Malawian health settings, co-facilitated the study workshops. We digitally recorded the overall group discussions for the second workshop using a tape recorder. LCSK and one participant took minutes in note pads and collected the flip charts from both workshops.

LCSK also engaged with the national officers between the workshops to discuss findings and seek approval to evaluate other initiatives agreed upon by stakeholders, such as additional variables in maternity registers to improve the identification of infants exposed to HIV at high risk.

### Data analysis

Descriptive demographic characteristics for stakeholders were analysed using Microsoft Excel and Stata 14 for the frequencies, proportions, and median (the number that divides the data set into two halves, 50% below and above it).

We used a reflexive thematic approach to analyse qualitative data [34]. LCSK familiarised herself with field notes from workshop 1(S1 Data) and deductively identified intervention functions and modes of delivery (S2 Table). LCSK compiled an outline of the intervention and identified and prioritised problems with their contributory factors from workshop one, which was reviewed by EL, JHZ, AM, DN, and AO and then verified and discussed at workshop two.

After workshop two, researchers further inductively open-coded the transcribed recorded data to understand participants' meanings regarding the diagnosis of the problems, development, and refinement of the intervention, comprehensively following Braun and Clarke's phases of the analytical process [35].

In the first phase, LCSK familiarised herself with the data, listened, transcribed verbatim and verified transcripts. Then, LCSK read all transcribed workshop 2 audio and field notes from workshop 1 several times. In the second phase, LCSK generated initial codes on printed word documents and continued open coding in Nvivo Pro 12. In the third phase, LCSK categorised and revised codes in line with their meanings and relationships to answer research questions. LCSK, AO and ND further reviewed the classified codes in the fourth phase, considering the dataset and the generated codes. Data interpretations were further examined by co-researcher stakeholders (JHZ, EL, AM and DN) who participated in the engagement workshops and all authors as a means of member checking. In phase five, LCSK named and defined the themes reviewed by all authors [35].

## Results

### Characteristics of identified stakeholders engaged in co-designing EEH intervention

We engaged 44 stakeholders at two workshops, with a median age of 35 years and an age range of (24-61 years), whose characteristics are described in Table 1. Eleven of the 17 participants from the initial workshop also attended the second workshop with 27 stakeholders. Most stakeholders, 31(70%), were from the public sector.

### Results overview

Stakeholders defined the main problems in EID services as inadequate enrolment of infants exposed to HIV in HIV care clinic at birth, HIV testing and counselling at six weeks and the lack of patient-centred integration of services. Five overarching thematic areas emerged:1)

**Table 1. Characteristics of stakeholders engaged in co-designing EEH intervention.**

| Type of workshop | Formative | Refinement | Total |
|---|---|---|---|
| Total stakeholders identified and engaged | 17 | 27 | 44 |
| Categories of stakeholders | | | |
| Stakeholders (public sector) | 12 (71%) | 19 (70%) | 31(70%) |
| 1. Managers | 2 (12%) | 4 (15%) | 6 (14%) |
| 2. Coordinators | 4 (24%) | 6 (22%) | 10 (23%) |
| 3. Service providers | 5 (29%) | 8 (30%) | 13 (30%) |
| 4. Researchers | 1 (6%) | 0 | 1 (2%) |
| 5. Policy | 0 | 1(3%) | 1 (3%) |
| Stakeholders (community and private sector) | 5 (29%) | 8 (30%) | 13 (30%) |
| 1. Service users | 2 (12%) | 2 (7%) | 4 (9%) |
| 2. Non-governmental organisation | 2 (12%) | 5 (19%) | 7 (15%) |
| 3. Policy | 1 (5%) | 1 (4%) | 2 (6%) |
| Median age | 35 (24-61) | 39 (32-61) | 39 (24-61) |
| Sex (Female) | 11 (65%) | 19 (70%) | 30 (68%) |
| Qualifications: | | | |
| 1. Primary | 2 | 2 | 4 |
| 2. Secondary | 1 | 3 | 4 |
| 3. Certificate | 3 | 5 | 8 |
| 4. Diploma | 4 | 5 | 9 |
| 5. Bachelor's degree (BSc) | 6 | 8 | 14 |
| 6. Master's degree (MSc) | 1 | 3 | 4 |
| 7. Doctorate (PhD) | 0 | 1 | 1 |

enabling client identification, 2) context-appropriate client-centred service integration, 3) enhancing coordination and accountability, 4) strengthening capacity building for optimal service delivery and 5) sustainability plans for the intervention. Sub-themes are categorised as contributory factors operating at different implementation levels classified in COM-B subdomains and co-designed initiatives to address challenges (Fig 3). Finally, the co-designed EEH intervention is summarised (Table 2 and Fig 4).

## Theme 1: Enabling client identification

Stakeholders reported barriers to client identification that affected infants exposed to HIV enrolment in HIV care clinics and HIV testing at six weeks in rural and urban facilities at client, provider and system levels as follows:

1. Client-level

   **Stigma.** To avoid discrimination from employers requiring health passports and to protect their privacy from friends and family, clients were hesitant to use health passport books that documented HIV-positive status. Some removed the HIV results page or used new books. *"They sometimes change books because they do not wish to be identified as living with HIV" (WK2HCW7).* Others provided false information, denied their status despite counselling, and stopped accessing services, obstructing the identification of infants exposed to HIV.

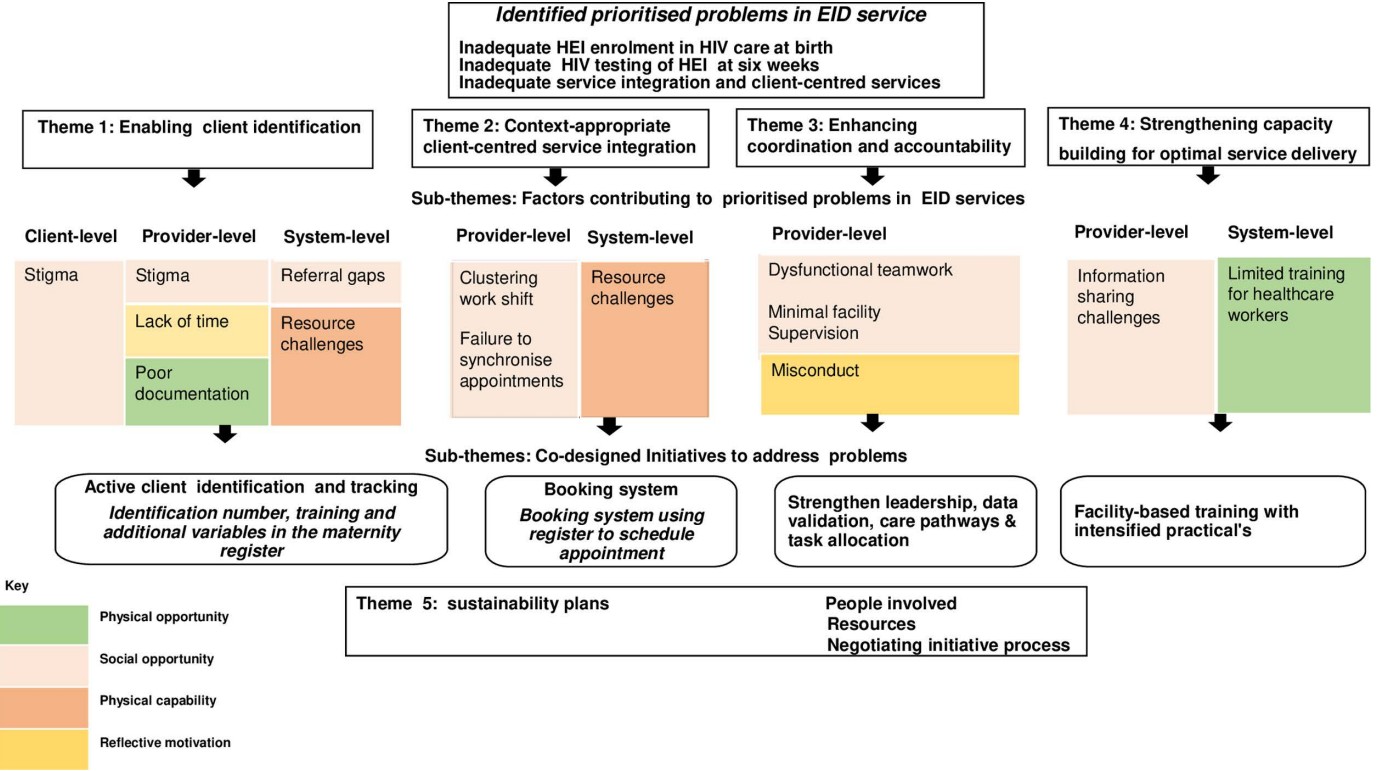

**Fig 3. Summary of themes and subtheme.**

2. Provider-level

**Impact of perceived stigma.** Provider awareness of the adverse effects of stigma led HCW to avoid checking clients' HIV status to prevent disclosure because services were offered in crowded places. Some feared women would abandon services like family planning if referred for HIV testing. Additionally, because testing was not mandatory, they struggled to identify infants exposed to HIV.

**Lack of time.** The focus on providing only one service to save time led to HCW not identifying infants exposed to HIV. *"Like us, health surveillance officers at the under-five, we are too quick and focus only on immunisation" (WK2 HCW group1).*

**Poor documentation.** HCW's failure to transfer HIV information from records to health passports and inaccuracies on monitoring cards hindered infant identification for follow-up testing.

3. System-level

**Resource challenges.** Maternal HIV testing rooms were congested and far from the labour ward because multiple departments tested clients in one location due to inadequate providers. Consequently, HCW often referred women for HIV testing after delivery or did not provide testing at birth, hindering the identification of infants exposed to HIV.

The lack of POC viral load testing for mothers at primary facilities posed challenges for HCW in identifying infants at high risk exposed to HIV. Women lacked viral load results at birth, and HCW relied on self-reported ART adherence. The available maternal data sources lacked variables to prompt HCW to assess infants at high-risk exposure.

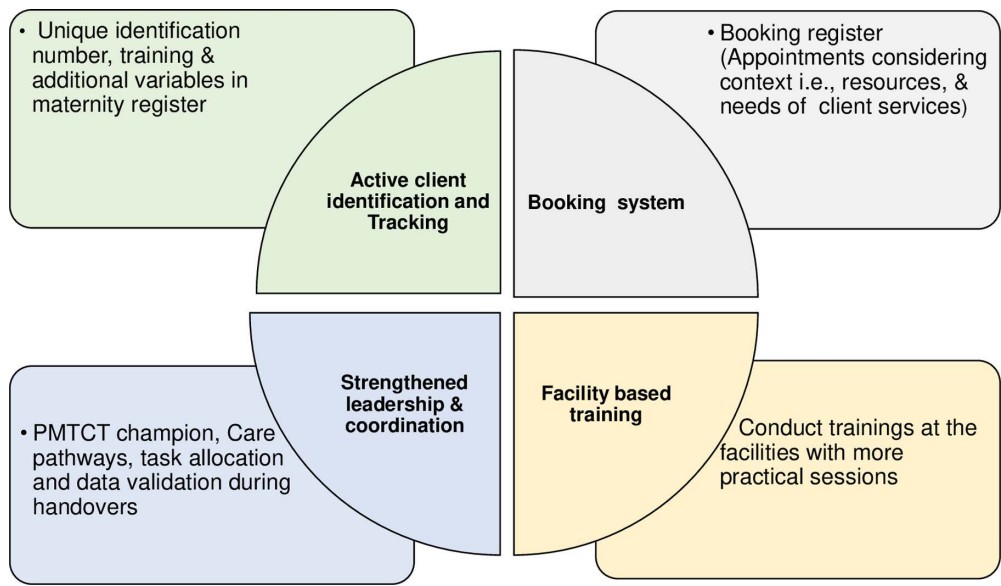

**Fig 4. Co-designed context-appropriate EEH intervention.**

**Referral gaps.** Women with delivery complications were often referred to a tertiary hospital without HIV testing. One stakeholder described a circumstance where a woman who was referred delivered twins and was again discharged at the tertiary facility without HIV testing. One baby later became seriously ill and tested HIV positive before dying. The mother denied the baby's HIV status, claiming she tested HIV-negative during pregnancy. *"The woman denied that the results were hers, as she indicated that she had tested HIV negative during her antenatal care. However, when we examined the documents, we found that the woman had not been offered an HIV test during the delivery period" (WK2HCW5).*

## Strategies to improve client identification and tracking

**Unique identification.** To address client identification barriers in Workshop 1, stakeholders proposed using a unique identification number instead of recording actual HIV-positive results. In workshop 2, some stakeholders felt this could be misinterpreted as discrimination. *"We need to discuss this further because the unique labels may lead to discrimination "(WK2 managers group 3).* Others suggested using unique identification only among HCW, ensuring confidentiality. Stakeholders discussed how unique identification could prevent HCW from inquiring about HIV status in crowded settings, saving time by avoiding referrals for clients testing who already tested positive but lacked results documentation. *"We note women without documented HIV status who already know they are HIV positive… and we refer them again to HTS; they stand in the line, which takes much time… if we have that identifier, we can track the women easily and save time "(WK2HCW3).* Stakeholders agreed on the unique identification description (Table 2) and considered electronic identification ideal. However, they decided to evaluate manual usage first due to limited resources. Additionally, they recommended standardising the unique identification value nationally to prevent confusion among healthcare workers and documenting it discreetly in the health passport, such as on the middle pages.

**Healthcare worker training and additional variables in the maternity register.** Stakeholders suggested training HCW to ensure confidentiality when using unique

**Table 2.  Description of the context-appropriate co-designed enhanced health system intervention.**

| Intervention Component | EEH intervention description | | | | | | |
|---|---|---|---|---|---|---|---|
| | Aim/why | What | How | Who | Where | When | Mode of delivery |
| Active client identification and Tracking | To identify and track infants exposed to HIV to enrol in an HIV care clinic and offer HIV testing at six weeks and maintain privacy. | Unique identification Number | Indicate identification number amidst other information in the health passport on the antenatal page. | Nurses, clinicians, HIV diagnostic assistants and health surveillance assistants | Antenatal, postnatal. Under-five, family planning clinic and labour ward and at every encounter | At every encounter, including birth, one and six weeks | In-person |
| | | | Check for identification or HIV positive or unknown status at each encounter; Do not ask clients for HIV status in a queue. | HCW and support staff | At any encounter | | |
| | To guide nurses in identifying infants at high risk to provide appropriate prophylaxis | Additional variables | Enter information for the mother and infant on the added variables and act according to the collected data. | Nurses | Labour ward | At birth | In-person |
| Booking system | To prompt nurses to reflect on contexts and clients' service needs to give appointments, to offer client-centred service integration, including HIV testing at six weeks | Booking register to schedule appointments and a guide to show the numbers booked and the agreed number of infants to book per visit | Enter clients' information and give appointment: variables will prompt reflections on context and clients' needs. Discuss the needs, expected services at six weeks' appointment and date of the service with clients. Book 3 clients on a day to accommodate the capacity of the POC machine and additional clients that may walk in from other facilities, for example, clients that delivered at referral hospital & had no booking | Nurses and clinicians | Labour ward & postnatal clinic | At birth, one week (those missed at birth and those who gave birth at another facility) | In-person (hard-copy register) Face to face |
| | | | Plan resources for six weeks, i.e. human resources and equipment | Nurses with clients | Labour ward/ postnatal clinic | At birth, one and six weeks | Face to face |
| | | | Provide all required services for a woman at one visit | All HCW | labour ward, family planning, lab, immunisation etc. | | |
| strengthening leadership, data validation, care pathways and task allocation | To improve coordination and accountability to improve enrolment of infants exposed to HIV in HIV care, HIV testing and service integration | Complement the EID focal person with a champion | Health centre management to appoint a champion in the prevention of vertical transmission programme | Nurse | Labour ward | | |
| | | Task allocation | Develop a schedule on paper indicating services to be offered, place and person. | HCW (champion, nurses, managers) | Health facility | Every Thursday to review before effective on Monday | Posted in the office and sharing through WhatsApp |
| | | Care pathways | Plan and discuss steps of care for clients to follow at the facility | HCW | Health facility | During intervention training and review progress at the facility | |
| | | | Plan services by discussing with clients of expected services for six weeks appointment | As above on the booking system | | | |

*(Continued)*

**Table 2.** (Continued)

| Intervention Component | EEH intervention description | | | | | | |
|---|---|---|---|---|---|---|---|
| | Aim/why | What | How | Who | Where | When | Mode of delivery |
| | | | Prioritisation clients in the prevention of vertical transmission programme | HCW | At any encounter | At birth and six weeks | In-person |
| | | | Escorting clients | HCW/ support staff | From the labour ward and any encounter | At six weeks | In-person |
| | | | Counselling before HIV testing | HIV testing services provider | Antiretroviral therapy clinic | At six weeks | Face to face |
| | | Data validation during handovers | Staff receiving handovers should review client documents from different data sources (maternity register, HIV care clinic register, and pink cards) while the staff finishes the duty reports. | Nurses, All HCW | Labour ward at each handover | Each handover when changing shift | Face to face |
| Facility based training | To enhance capacity building for optimal service delivery | Facility-based training with more practice | Conducting training at the facility, including sessions to practice | All HCW | Health facilities or places close to the facility | When there are new guidelines, lessons mentorship and EEH intervention training | Face to face |

Key: co-design an enhanced health system context-appropriate intervention (EEH), healthcare workers (HCW), point of care (POC)

identifiers to prevent discrimination. To better identify infants at high risk and provide suitable prophylaxis, they discussed adding variables like maternal viral load and medication adherence to maternity registers, which were absent (S4Text).

## Theme 2: Context -appropriate client-centred service integration

Barriers to service integration were considered at provider and system levels as follows:

1. Provider-level

**Clustering work shifts.** Stakeholders discussed that HCW clustered work shifts to extend off-duty periods at rural facilities, leading to high workloads and insufficient services for clients at appointments.

2. System-level

**Resources challenges.** Urban and rural facilities had specific clinic days for mothers and infants due to limited infrastructure and few HCW. *"Our challenges are that some services at our mother and infant-specific clinic are unavailable. A woman who comes requiring family planning methods…may need a consultation with the doctor; the infant may require HIV testing… But HCW will only provide ARVs and check viral load…." (WK2NGO group2).* HCW recognised service integration's importance, but clients could not access all necessary clinic services due to HCW shortages, dispersed services, and the limited capacity of POC machines. Stakeholders described having few POC machines in the district, which process one sample at a time for up to an hour. A positive HIV result requires HCW to repeat the test, extending the testing duration for an infant to up to two hours. *"We only have four sites with POC out of 28 facilities …. If the infant is tested HIV positive, they repeat the test …." (WK2manager3)*

Additionally, cartridge shortages and POC machine malfunctions hindered HIV testing availability for infants exposed to HIV at six weeks.

Stakeholders noted that the absence of diverse programme mentors during integrated service supervision and a lengthy checklist negatively impacted service supervision and integration improvement.

## Strategies to improve integration

**Booking system.** In workshop 1, stakeholders developed an appointment booking system to help HCW schedule client appointments, addressing barriers to client-centred service integration and HIV testing. The booking system aimed to help HCW consider clients' needs and context, including POC machine capacity and tests processed during working hours at the urban facility. At workshop 2, stakeholders refined that the booking system should guide both facilities with and without POC machine for HCW to reflect on the number of women with HIV in the catchment that access HIV testing for infants exposed to HIV, specific clinic days if the facility could not provide daily services, availability of HCW, available resources and clients required services.

Views varied on the booking system platform, including manual versus electronic registers. Electronic booking was initially proposed for accuracy and speed. "*…the challenge that may arise is doing this manually. There may be a mix-up of dates… unless we do this electronically, the system can note that the date is full" (WK2NGO1).* Due to limited resource contexts, stakeholders created hardcopy registers (S5 Text) to accommodate three booked clients and space for two walk-ins from other facilities and recommended opting for DBS collection to avoid sending back clients. "*My worry is if we put five and send back two. We should consider where they are coming from; sending the two back may not be good. Collecting DBS and having them come another time for the results is important" (WK2 NGO3)*

**Healthcare workers' training and mentorship to support the booking system.** Stakeholders anticipated that HCW might confuse appointment dates using a manual booking system and struggle to register infants exposed to HIV if multiple birth and one-week postnatal care locations existed. They recommended training, ongoing meetings, and mentorship to resolve these issues.

## Theme 3: Enhancing coordination and accountability

Barriers to coordination and accountability were identified as follows:

1.  Provider-level

**Dysfunctional teamwork and minimal facility supervision.** Stakeholders reported poor teamwork and role segregation among HCW, leading to low birth enrolment in HIV care clinics. For instance, nurses left infants exposed to HIV enrolment responsibilities to health surveillance officers at six weeks instead of at birth, worsened by minimal supervision and the unavailability of health surveillance officers in maternity wards.

**Inadequate care coordination.** Stakeholders discussed that unclear care pathways led to healthcare workers providing insufficient client service information. Consequently, infants exposed to HIV began accessing POC HIV testing late, resulting in long waiting times for infant results. This reduced daily POC HIV tests and limited access to immunisation and family planning for mothers and infants. "*We go home very late at around one or twelve noon despite going early to the hospital…They ask us to sit somewhere, and we don't know why. (WK2 women2).*

**Misconduct.** Stakeholders reported HCW misconduct, including coming to work drunk and leaving women unattended during delivery. "*We are unsure where HCW spend their night*

*before coming to work. Usually, they come to work drunk and shout at us"* (WK2 Women group 4). Some HCW skip enrolling infants in HIV care, citing high workload when approached by leaders. *"…even when you ask them, they will express that they were busy even when it seems not to have been busy…" (WK2HCW3).*

### Initiatives to strengthen leadership, care pathways and task allocation

**Strengthened leadership.** To address supervision and teamwork barriers, stakeholders suggested facilities identify a champion (nurse or midwife) in the maternity ward to support the EID health surveillance focal person and enhance infants exposed to HIV enrolment at birth. Stakeholders discussed the need for collaboration for focal persons with the health facility management team.

**Data validation.** Stakeholders believed that data validation could improve accountability and support to address documentation and care gaps. *"…HCW should be tracked and answerable. If we can develop that mechanism, we may help people avoid shortcuts in their work". (WK2NGO2).* Stakeholders discussed improving accountability by having HCW validate data daily during handovers by reviewing client documents, focusing on women with HIV, those with unknown status, and infants at high risk and enrolment in the HIV care clinic. *"… through handovers during reporting…the nurses allocated to the post-natal or labour ward should verify documents" (WK2HCWgroup1).*

**Care pathways and task allocation.** Stakeholders suggested that HCW develop care pathways aligning with their context to improve service coordination with EEH intervention considering the following. Informing women in advance during birth discharge and at one week of the necessary services at six weeks and encouraging them to come early and meet nurses directly without queuing. *"…before providing post-natal care, the nurse needs to identify these women, and that unique identification needs to be included, and the woman needs to be given the information." (WK2HCW7)*

Support staff weighing infants should separate health passports with unique identification and without documented HIV status to give nurses first. In a private space, HCW must assess clients and provide or refer them to HIV testing services to prevent disclosing their HIV status in overcrowded environments. Prioritise post-natal services for women with HIV to ensure early referral to POC testing and accommodate longer processing times. Escort mother and infant to receive required services during sample processing. Contested views emerged regarding escorting clients between service points to reduce waiting times and improve service access. Some argued that such escorting could unintentionally disclose women's HIV status, and support staff might not always be available for this task.

Stakeholders from urban facilities believed escorting would help clients access services, navigate the setting with fragmented infrastructure, and avoid delays. They agreed that all HCW should understand their roles in task allocation. Rural stakeholders preferred to provide certain services in a single room. Considering their context, such as infrastructure and few clients, they preferred having various HCW come to the single room rather than having clients move around.

### Theme 4: Strengthening capacity building for optimal service delivery

Barriers to coordinated capacity building were identified as follows

1.  System-level

**Limited training for healthcare workers.** Stakeholders stated that most HCW lack training on new guidelines due to limitations in funding. *"Very few people have been trained*

*because of limited funding."* (WK2 manager3). The training for integrated elimination of vertical transmission/ ART/ Tuberculosis guidelines in Malawi targets clinicians, nurses, and midwives with few health surveillance officers who mainly provide EID services. Additionally, training for HIV testing services focuses on health surveillance officers and diagnostic assistants, while lab technicians providing EID with POC are untrained and offer the service without counselling the mother and infant.

2. Provider-level

**Information sharing challenges.** Stakeholders noted that poor information sharing among HCW leads to knowledge gaps. Some HCW withhold their insights and neglect to educate students during clinical practice due to insufficient understanding during their training. *" Some people that went for training do not fully understand the content. When back, they do not share the information because they are not sure if they will share the correct information" (Wk2NGOgroup2).* Some HCW rely on non-governmental organisations to mentor HCW who have never attended the training. Some HCW reject information from trained peers due to a lack of training allowances. *"…if I have gone for training, even if I inform my friends, they will say he went to eat the money alone 45 000 Malawi Kwacha…"* Wk2HCWgroup1)

### Initiatives to address the barriers to HCW knowledge

**Facility-based training with more practical skills.** Stakeholders recommended facility-based training to maximise limited funding to train more HCW while reducing night accommodation costs. To ensure equal access to lunch allowances, to motivate all HCW. For learning materials, stakeholders suggested integrating more practical sessions to enhance skill acquisition, prioritising program-specific goals before including service integration and training all cadres for better coordination. Therefore, stakeholders advocated training all HCW cadres for the EEH intervention using facility-based training.

> *. …"Apart from having training integrating …ART, EID, ETC, can't we still consider having programme-specific training? Targeting support staff who are our backbone in implementation… through the facility-based training?" (WK2 manager 3)*

### Theme 5: Sustainability plans for the EEH intervention

Stakeholders discussed sustaining the co-designed initiatives in the following key areas:

**People involved.** Good leadership, champions, quality improvement teams, HCW commitment, and community involvement were crucial factors discussed by stakeholders to sustain the EEH initiative.

Stakeholders identified good leadership as vital for sustaining initiatives and emphasised the need for a champion to lead implementation with management support. The champion's role includes leading problem-solving for the intervention and communicating decisions to HCW for guidance.

Facilities must implement quality improvement initiatives like client identification and care pathways, monitor progress, and address challenges. Stakeholders linked good leadership to HCW commitment, indicating that leaders should provide HCW equal training opportunities for teamwork. *"The commitment of HCW can be achieved if they are well supported—for example…The leaders must be inclusive, considering all HCW rather than inviting the focal persons alone to meetings." (WK2NGO group2)*

Stakeholders suggested community training on the importance of services to boost access and compliance with medical advice.

**Resources.**  Transport for mentorship support and supplies like booking registers were identified as essential resources. They acknowledged existing non-governmental partnerships that could assist with transportation, and some partners were ready to help district managers reach facilities for mentorship. '*We can support transport for mentorship*"*(WK2 NGO4).*

**Negotiating initiative process with healthcare workers.**  Intervention benefits, training, and mentorship were factors stakeholders thought could encourage HCW buy-in for sustainable implementation and felt the intervention was essential to address the problems. "*We aim to cover 100% in identifying clients. We do not want to miss any infant exposed to HIV. We miss the clients, but with the* intervention, we *want to identify and track every infant…*" *(Wk2 managers group 3)*

Training staff to comprehend and apply the intervention and its benefits was deemed essential for sustainability, along with continuous mentorship and the orientation of new HCW.

## Final co-designed EEH intervention

The Co-designed EEH intervention (Fig 4) has four broad components:

Active client identification: HCW checks and indicates in health passport books unique identification numbers to track clients. Adds variables to maternity registers to identify infants at high risk for appropriate prophylaxis and training HCW to use identification respectfully while maintaining client dignity and privacy.

The booking system register prompts HCW in all facilities to assess context and client needs for scheduling appointments, ensuring client-centred service integration and comprehensive service delivery.

Strengthening leadership, data validation and care pathways. Strong leadership involves appointing a champion person to lead service implementation in maternity. Data validation ensures the accuracy of daily reports from HCW handovers. Care pathways improve care flow, prioritise clients, ensure proper escorting, centralise services, and facilitate discussions with clients, promoting HCW coordination, timely delivery, and accountability.

Facility-based training offers practical training within the facility to enhance HCW skills and promote teamwork.

## Discussion

Low utilisation of health services after birth continues to affect early infant diagnosis of HIV, endangering the aim to end AIDS by 2030. Collaborative research with stakeholders to understand the context and co-design interventions to address suboptimal EID services is limited. Our study engaged stakeholders and acknowledged power dynamics, navigating small homogenous group discussions during workshops to enhance collaboration.

Stakeholders analysed context-specific health system EID service challenges, found EID services to be sub-optimal, and co-designed a context-appropriate EEH intervention emphasising sustainability. Challenges identified and respective interventions fell into five themes: (1) enabling client identification, (2) context-appropriate client-centred service integration, (3) enhancing coordination and accountability, (4) capacity building for optimal service delivery, and (5) sustainability plans for the intervention. The co-designed context-appropriate intervention comprised active client identification and tracking, a booking system, strengthened leadership, data validation, care pathways, task allocation and facility-based training.

We classified the challenges for each theme into COM-B subdomains (Fig 3). Although the factors varied, those related to social opportunity predominantly appeared in three themes [30], such as referral gaps affecting client identification, clustering work shifts affecting client-centred service integration and dysfunctional teamwork affecting coordination and accountability of HCW. Physical opportunity factors, such as fragmented and crowded service delivery spaces, were cross-cutting, affecting client identification and service integration. Reflective motivation factors reinforced each other across themes, such as limited time for HCW to check HIV status led to a lack of patient identification and, subsequently, a lack of integrated services. Physical capability factors further impeded client identification and limited the capacity of HCW to provide EID service. These findings confirmed Suwedi-Kapesa et al. findings and clarified contextual factors for suboptimal EID services [18].

Stigma manifesting at client and provider levels hindered client identification and raised privacy concerns about HIV status to prevent discrimination and exclusion [36]. Lack of privacy has led to loss of follow-up in prevention of vertical transmission programmes and prevented HIV testing in adults [36–38]. In our study, client-level stigma was anticipated, creating a missed opportunity for mother and infant to access comprehensive services [36]. At the provider level, anticipated stigma compelled by system-level factors compromised HCW to check HIV status for clients as recommended by the guidelines in Malawi [39,40]. According to Levesque et al. [41], access is described as achieving the required healthcare needs [41]. Therefore, multifaceted stigma factors in our study compromised women with infants exposed to HIV from accessing comprehensive services when already present at health facilities.

Unique identifiers evolved in workshops to improve client identification amidst stigma factors. Past research recommended against inquiring about clients' HIV status in crowded settings, advocating for documenting their actual status to prevent inadvertent disclosure and discrimination, respectively [36,42]. Similarly, the WHO recommends using unique identification to repeatedly and correctly identify clients for comprehensive services [43].

Our findings suggest gaps in comprehensive person-centred care services due to provider and system-level factors [44]. For example, the POC machine's limited capacity, HCW failure to synchronise appointments and clinic-specific days prevented women with infants exposed to HIV from accessing required services [44]. Suwedi-Kapesa et al. and previous studies have also reported these factors [12,18]. Our findings add that delaying HIV testing in infants using POC machines decreases access to testing. This indicates a need for better time management, increased POC machines, or innovations to improve POC capacity to meet WHO guidelines [8]. POC testing is effective, but understanding the context of its implementation is key. Findings suggest a booking system to optimise service integration rather than triaging clients at appointments, which previously led to infants exposed to HIV leaving without HIV testing [18]. The booking system could further alert HCW of women with infants exposed to HIV who miss appointments for early tracing [2].

Planning the daily number of clients to be tested according to the capacity of the POC machine is consistent with the UNICEF lessons to guide the scaling up of POC testing [45]. Different programmes have shown that the booking system addresses waiting times [46,47]. Our findings add the importance of guiding HCW to reflect on their context, plan coordination, and integrate services.

Although our study site selection reflected the context of location (urban versus rural), presence or absence of POC machines, and volume of clients served at the facility. Our findings suggest that several contextual aspects interact and affect each other, limiting integrated services, client identification, capacity for HCW and coordination to achieve intended outcomes. Key contextual factors included the daily tests that the POC machine processes,

staffing levels, crowded spaces, specific clinic days, limited resources, incentives for training attendance, lack of care pathways, and clients' required services.

In workshops, urban and rural facilities reflected differently on how integrated services would manifest depending on infrastructure, client numbers, POC availability, and staffing. While both agreed that one-stop integration wasn't feasible, rural stakeholders suggested combining certain services in one room instead of clients moving around, which the urban facility did not find practical based on their infrastructures. Previous studies have noted various service integration models [48–50], and some have not been effective [49,51], suggesting the need for researchers to embrace context-responsive research approaches to enhance reflection of context and implementation of context-responsive interventions [52].

A booking system that includes key contextual variables to meet integration guidelines can help HCW schedule appointments and unique numbers may enhance client identification in crowded areas, potentially improving service uptake. Nevertheless, the booking system may be challenged by a lack of adherence and emergencies where clients would come without or miss appointments. Our findings provide possible measures to address such challenges. As recommended by Suwedi Kapesas' et al [18], we emphasise that HCW should collect dry blood spot samples for near point-of-care testing when POC testing isn't available.

Our findings indicate unclear roles in EID services, such as infant enrolment in HIV care clinics among HCW and at different levels, despite existing standard operating procedures. Additionally, the clustering of work shifts affects HCW accountability. Agreeing with Cleary et al., unclear roles and task allocation likely impact accountability [39,53]. Our findings on the care pathways initiative align with the UNICEF recommendation to optimise diagnostic networking for scaling up WHO-recommended POC machines' use [45]. Findings suggest data validation at handovers and strengthening leadership can enhance accountability and promote internal audits for better health outcomes among colleagues [53].

Consistent with other programmes, information sharing among HCW is affected by the reluctance to receive information and work as a team by those without access to training because of lack of training allowances, which HCW considers an incentive [54]. The findings indicate that HCW training may not meet its intended outcomes due to inadequate understanding and training few HCW. There is a need for more facility-based training that includes practical sessions to enhance skills, as supported by a previous study highlighting the effectiveness of onsite training with ongoing supervision [55]. We recommend evaluating current teaching methods, logistics, and the feasibility of facility-based training, mindful of infrastructure challenges. Additionally, exploring training venues near facilities may help train more HCW with fewer allowances, fostering teamwork.

Our study's limitations included excluding training institutions, with students lacking knowledge and support at health facilities. Stakeholders discussed strategies to improve HCW knowledge for better learner support. Nevertheless, engaging various public, community and private stakeholders from different catchments and health system levels at workshops provided a rich understanding of our study objectives [18,19,25]. To manage power dynamics, we had small group discussions before entire group discussions, which may have restricted views brought to the larger group. However, a broader discussion highlighting diverse opinions followed each group's consensus presentation. The group discussions used BCW intervention development prompts Fig 2, but inductive questions arose from the shared experiences in small and overall group conversations. Engaged stakeholders from five of 28 primary facilities may limit understanding in Blantyre, but we also involved program managers who oversee all 28 facilities and the referral hospital who shared comprehensive experiences. Our context-focused approach might restrict replicating findings. Still, our results highlight essential methodological aspects for co-designing interventions and key areas necessary for effective service delivery, like client identification, person-centred service integration, and leadership.

Our study offers several lessons. Consultation and networking are essential for identifying stakeholders to engage unknown to researchers. Co-design takes time before, during, and after workshops to develop and modify interventions. Researchers must establish clear objectives to prompt discussion by applying theories and be open to initiating follow-up questions, as workshops reveal insights that may not occur to researchers that address contextual needs. Researchers must develop interpersonal skills, understand power dynamics, and educate stakeholders on co-design theories to identify key problems after exploring their experiences and co-design interventions. Combining frameworks for our theory broadened our understanding of essential factors in co-designing health system interventions, such as context, sustainability and behaviour change cues. Stakeholder engagement is critical for ownership, sustainability, and policy influence [26,56]. The uncertainties of the intervention will be assessed during the intervention evaluation.

We acknowledge common EID approaches focusing on evidence-based implementation science research. However, due to inconsistent intervention effectiveness in various settings [15,48,57], we recommend researchers use co-design methods to create context-specific strategies. Policymakers should insert prompts in HIV service guidelines to assist facilities in reflecting on their context on how to meet requirements, such as client identification and person-centred service integration. Additionally, define staff roles clearly and implement measures like leadership and handovers to enhance HCW accountability. Evaluate training aligned with their context to maximise resources in improving HCW capacity and teamwork.

## Conclusion

We found inadequate infant enrolment in HIV care clinics, HIV testing and service integration as the main problems in EID service implementation compounded by complexities in client identification, inadequate patient-centred integration of services, coordination, accountability and HCW knowledge. Unique identifiers, booking systems, strengthening leadership, data validation, care pathways, and facility-based training were the main aspects of the co-designed context-appropriate intervention to improve the uptake of EID services. These findings provide stakeholders' co-designed context-appropriate initiatives to address suboptimal EID services. Nevertheless, they address problems common in other low-resource settings in EID services. They propose ways of maximising limited POC testing machines to achieve WHO recommendations for POC testing in infants exposed to HIV and integrate services reflecting on context. We recommend evaluating the co-designed initiatives in addressing the identified EID service implementation gaps.

## Supporting information

**S1 Text. Inclusivity in global research questionnaire.**
(DOCX)

**S2 Text. Summary of previous study findings.**
(PDF)

**S3 Text. Summarised findings from Formative Workshop 1.**
(PDF)

**S4 Text. Additional page of variables for register.**
(DOCX)

**S5 Text. Booking system register.**
(PDF)

**S1 Table. Intervention functions and modes of delivery.**
(XLSX)

**S2 Table. Characteristics of engaged public facilities.**
(XLSX)

**S1 Data. Workshops notes and transcripts.**
(ZIP)

## Acknowledgment

This paper is part of PhD studies for the first Author at the Liverpool School of Tropical Medicine at the Department of International Public Health. We express gratitude to all stakeholders engaged in the workshops. Thanks go to the Blantyre district health office and all management teams for the local and international organisations that released representatives to participate in the workshops in the Blantyre district. We are thankful for the support from Policy experts at the Malawi Liverpool Wellcome Trust, Kamuzu University of Health Sciences, College of Medicine, Health Systems and Policy and Malawi's Ministry of Health Directorate of HIV, STI and Viral Hepatitis. Special thanks go to Professor Miriam Taegtmeyer and Dr Helen Nabwera for their guidance on the concept development.

## Author contributions

**Data curation:** Leticia Chimwemwe Suwedi-Kapesa.

**Formal analysis:** Leticia Chimwemwe Suwedi-Kapesa.

**Funding acquisition:** Leticia Chimwemwe Suwedi-Kapesa.

**Investigation:** Leticia Chimwemwe Suwedi-Kapesa, Jenifer Hezekiah Zimba, Edda Lipipa, Dorcus Nothale, Afunawo Mdala.

**Methodology:** Leticia Chimwemwe Suwedi-Kapesa.

**Project administration:** Leticia Chimwemwe Suwedi-Kapesa, Augustine Choko, Linda Alinane Nyondo-Mipando, Jenifer Hezekiah Zimba, Edda Lipipa, Dorcus Nothale, Melody Sakala.

**Resources:** Leticia Chimwemwe Suwedi-Kapesa.

**Software:** Leticia Chimwemwe Suwedi-Kapesa.

**Supervision:** Leticia Chimwemwe Suwedi-Kapesa, Augustine Choko, Linda Alinane Nyondo-Mipando, Melody Sakala, Nicola Desmond, Angela Obasi.

**Validation:** Leticia Chimwemwe Suwedi-Kapesa, Jenifer Hezekiah Zimba, Edda Lipipa, Dorcus Nothale, Afunawo Mdala, Joe Nkhonjera, Nicola Desmond, Angela Obasi.

**Visualization:** Leticia Chimwemwe Suwedi-Kapesa.

**Writing – original draft:** Leticia Chimwemwe Suwedi-Kapesa.

**Writing – review & editing:** Leticia Chimwemwe Suwedi-Kapesa, Augustine Choko, Linda Alinane Nyondo-Mipando, Jenifer Hezekiah Zimba, Edda Lipipa, Dorcus Nothale, Afunawo Mdala, Joe Nkhonjera, Melody Sakala, Nicola Desmond, Angela Obasi.

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
