## [Decision Letter · Decision Letter 0]

3 Jan 2025

PGPH-D-24-02616

Developing an intervention to improve early infant HIV diagnosis service uptake among postpartum women in Malawi’s primary healthcare using a co-designing approach with stakeholders

Dear Dr. Suwedi-Kapesa,

Thank you for submitting your manuscript to PLOS Global Public Health. After careful consideration, we feel that it has merit but does not fully meet PLOS Global Public Health’s publication criteria as it currently stands. Therefore, we invite you to submit a revised version of the manuscript that addresses the points raised during the review process.

We look forward to receiving your revised manuscript.

Kind regards,

Joel Msafiri Francis, MD, MS, PhD

Academic Editor

Journal Requirements:

2. In the ethics statement in the Methods, you have specified that verbal consent was obtained. Please provide additional details regarding how this consent was documented and witnessed, and state whether this was approved by the IRB.

3. Please amend your detailed Financial Disclosure statement. This is published with the article. It must therefore be completed in full sentences and contain the exact wording you wish to be published.

**Please only choose the relevant sentences from below**

a. Please clarify all sources of funding (financial or material support) for your study. List the grants (with grant number) or organizations (with url) that supported your study, including funding received from your institution. 

b. State the initials, alongside each funding source, of each author to receive each grant.

c. State what role the funders took in the study. If the funders had no role in your study, please state: “The funders had no role in study design, data collection and analysis, decision to publish, or preparation of the manuscript.”

d. If any authors received a salary from any of your funders, please state which authors and which funders.

4. We note that your Data Availability Statement is currently as follows: "The datasets used and or analysed during the current study are all included in the manuscript as part of the results or supporting file."

5. Please insert an Ethics Statement at the beginning of your Methods section, under a subheading 'Ethics Statement'.

6. Please provide an Author Summary. This should appear in your manuscript between the Abstract (if applicable) and the Introduction, and should be 150–200 words long. The aim should be to make your findings accessible to a wide audience that includes both scientists and non-scientists. Sample summaries can be found on our website under Submission Guidelines:

https://journals.plos.org/globalpublichealth/s/submission-guidelines#loc-parts-of-a-submission

Additional Editor Comments (if provided):

Reviewers' comments:

Reviewer's Responses to Questions

**Comments to the Author**

1. Does this manuscript meet PLOS Global Public Health’s publication criteria ? Is the manuscript technically sound, and do the data support the conclusions? The manuscript must describe methodologically and ethically rigorous research with conclusions that are appropriately drawn based on the data presented.

Reviewer #1: Yes

Reviewer #2: Yes

2. Has the statistical analysis been performed appropriately and rigorously?

Reviewer #1: No

Reviewer #2: N/A

3. Have the authors made all data underlying the findings in their manuscript fully available (please refer to the Data Availability Statement at the start of the manuscript PDF file)?

Reviewer #1: Yes

Reviewer #2: Yes

4. Is the manuscript presented in an intelligible fashion and written in standard English?

Reviewer #1: Yes

Reviewer #2: Yes

5. Review Comments to the Author

Reviewer #1: The manuscript describes an important topic in relation to early infant diagnosis of HIV. Detailed information was provided on how this study was conducted. A few comments

1. Introduction could be strengthened with data on HEI cascade within Malawian contexts. Such as Numbers of HEI born at a particular time frame. Of those, what proportion were brought in for testing (say at 6 weeks). Of those what proportion got tested. Of those, what proportion received results. This information could set a stage to see the gaps the authors are trying to explain they do exist. This data was not provided besides qualitatively explaining existing gaps. Quantifying the magnitude of the problem would be beneficial.

2. Did Malawi adopt the WHO EID testing, or it has its own guidelines? How is the testing scheduled? Do the authors considers all testing before 18 month and EID? Or there is a time limit this is being referred? Defining it somewhere (introduction or methods) would be helpful

3. Line 74-76,There is a need to disentangle testing by age. If I am not mistaken PCR is as stated 4-6 weeks, then at 9months on is antibody testing. If the intention was to only consider PCR, there was no need to use 18- months threshold, else states other testing modalities after 6 weeks.

4. 4 POC in 28 facilities is difficult to achieve effective infant HIV testing. Compounded by the time it take to test single sample. What did not come out during stake holder’s discussions is the use of dry blood spot and sending samples to central lab. Would you consider this as option in your recommendations? How feasible is this?

5. Stating HIV positive women as in line 78 is considered stigmatizing. Can change to women living with HIV throughout the manuscript.

6. Analysis section was all about qualitative aspect of the project. Table 1, however, has quantitative data. It was not described in the analysis section. For example, while it is intuitive to think the numbers reported after median are lower and upper quartiles, not every reader is aware of. It would be appropriate to detail analysis section and document in the table what do those numbers mean.

7. A little difficult to follow certain information. For example, supplemental file S2, what do those H, Y, N, LDL mean? Having a footnote would be helpful.

8. If formative research was aimed at validating validate Suwedi-Kapesa’s work. Do you want to say something about this validation in your discussion or at least concluding remarks?

Reviewer #2: This interesting research study uses co-design to identify context-specific determinants and design appropriate implementation strategies. I enjoyed reading and learning from your manuscript. Below are some comments and suggestions for improvement.

Abstract

Overall, the abstract does not entirely clarify the context of the collaboration mentioned, the details of the setting, or the stakeholders. Please add this information to help the reader better understand your study.

Line 31: To be more apparent to the reader, please specify who is underutilizing health services after birth.

Line 32-34: Clarify the context in which this study was conducted. Why did this collaboration happen( is it part of an ongoing study?), who are the stakeholders, and what kind of EID services in Blantyre (in health facilities?, public/private?, and how many)?

Lines 41-46: When specifying the challenges, clarify the subjects to whom these challenges apply. For instance, who suffers from stigma or has clustered work shifts and minimal supervision? It would be a mistake to assume the readers can guess these.

Line 46: the abbreviation HCW is not defined anywhere in the abstract; please correct it.

Introduction

Overall, reduce the number of abbreviations to provide a smooth reading experience.

I recommend adding data about EID coverage in the study setting.

Line 66: it’s unclear if the estimated 72.3% and 82% are related to postpartum or ART care retention. Please clarify.

Line 82: define MRC

Line 76-86. The information in the paragraphs suggests that the present manuscript responds to a previous study’s findings. If that is the case, this should be clearly stated so the reader can understand the context of your study.

Material and methods

Overall, the content needs to be reorganized. I recommend first describing the program/intervention and clarifying the study setting and design. The results section is too long; it would benefit the reader if the information could be further synthesized. I couldn’t read the content of the figures in PDF format; the quality was poor.

Lines 92-94 read: “We co-designed the intervention to be evaluated at one urban and one rural study site, selected based on location, catchment population, and the presence or absence of POC machines.”

I don’t understand this text. Does “we co-designed the intervention to be evaluated” means “our study was conducted”? The way it is written, it seems that this co-designed intervention will( in the future) be evaluated at one urban and one rural study site. Were the workshops conducted at these two sites? Are the stakeholders from these two sites? Please clarify the setting of the present study.

Lines 93-97: How many urban and rural primary health facilities exist in Blantyre? You mentioned that they were selected based on location, catchment population, and the presence or absence of POC. Please explain more about these selection criteria so we can understand how you chose your study settings.

Lines 105-118: I am confused. It states, “ Our program theory…”. What program has been referred? I suggest describing the program first and then describing the theory. If your “program” is the same as your “study,” I suggest harmonizing the language to avoid confusion.

Line-120:123. In this section, please describe only the study designs. For instance, a descriptive study design of the stakeholder analysis content and a qualitative study design for the stakeholder’s discussion are used. Information about the intervention steps should be removed and added to a section describing the intervention/program.

Lines 126-141: The transition to the “recruitment and enrolment of stakeholders” section is weird because the workshop and its participants were not previously described in the paper. I recommend creating a section at the beginning of the methods section to describe the program/intervention(which seems to me to be the workshops) so that when we read the setting, study design, and all other sections, it is clear what we are talking about.

Lines 154-167: I recommend moving the information about the workshops' characteristics to the new program description section I suggested above. In the data collection section, please describe when the data was collected, who collected it, the instruments used, and the data types collected.

Lines 169-193: Again, this is part of program description. I recommend moving this information.

Line 240: I suggest using the exact wording as lines 224-227 to facilitate the information flow. Instead of “client identification,” I suggest writing “Theme 1: enabling client identification”. The same goes for the other four themes.

The results section is too long. Please summarize it as much as possible.

Discussion

Lines 582-588: As mentioned above, the selection of study sites needs to be clarified. I am confused about what you identify as study sites, the description of the workshops, and the discussion surrounding these sites.

Line 586-594: By reading these lines, it seems that the EEH was tested in the two study sites. Is this the case? Or have these conclusions been extracted from the participants of the workshops? Please clarify.

The only study limitation cited in your discussion is the exclusion of one group of stakeholders. I encourage the authors to consider other possible limitations to conducting such a study.

It would be essential to focus your discussion on the need for researchers to use co-design approaches to design context-specific implementation strategies. Besides the findings drawn from the workshop participants, what have you learned as a researcher when applying a co-design approach? What advantages and disadvantages can be pointed out?

What implications do your findings have from a research and policy perspective in Malawi?

6. PLOS authors have the option to publish the peer review history of their article (what does this mean? ). If published, this will include your full peer review and any attached files.

**Do you want your identity to be public for this peer review?** For information about this choice, including consent withdrawal, please see our Privacy Policy .

Reviewer #1: No

Reviewer #2: **Yes: ** Aneth Dinis

---

## [Editor Report · Decision Letter 1]

19 Mar 2025

Developing an intervention to improve early infant HIV diagnosis service uptake among postpartum women in Malawi’s primary healthcare using a co-designing approach with stakeholders

PGPH-D-24-02616R1

Dear Ms Suwedi-Kapesa,

We are pleased to inform you that your manuscript 'Developing an intervention to improve early infant HIV diagnosis service uptake among postpartum women in Malawi’s primary healthcare using a co-designing approach with stakeholders' has been provisionally accepted for publication in PLOS Global Public Health.

Best regards,

Joel Msafiri Francis, MD, MS, PhD

Academic Editor